# Adult Hepatitis B Virus Vaccination Coverage in China from 2011 to 2021: A Systematic Review

**DOI:** 10.3390/vaccines10060900

**Published:** 2022-06-06

**Authors:** Xinxin Bai, Lu Chen, Xinyao Liu, Yujia Tong, Lu Wang, Minru Zhou, Yanming Li, Guangyu Hu

**Affiliations:** 1School of Public Health, Hebei Medical University, Shijiazhuang 050011, China; 20202184@stu.hebmu.edu.cn; 2Department of Radiology, Wuhan Jinyintan Hospital, Tongji Medical College of Huazhong University of Science and Technology, Wuhan 430023, China; chenlu@whfjydyy999.onexmail.com; 3Institute of Medical Information, Chinese Academy of Medical Sciences and Peking Union Medical College, Beijing 100020, China; liu.xinyao@imicams.ac.cn (X.L.); tong.yujia@imicams.ac.cn (Y.T.); 4Beijing You’an Hospital, Capital Medical University, Beijing 100069, China; youan2@wjw.beijing.gov.cn; 5Qinghai Center for Disease Prevention and Control, Xining 810007, China; zhoumr@qhcdc.org.cn (M.Z.); liym@qhcdc.org.cn (Y.L.)

**Keywords:** hepatitis B virus, vaccination, adult, meta-analysis, China

## Abstract

Background: The most effective way to prevent hepatitis B virus (HBV) infection is vaccination. Synthesized data on vaccination coverage in adults against hepatitis B in China are scarce. We aimed to estimate the hepatitis B vaccination rate in adults in China. Methods: We searched PubMed, EMBASE, Cochrane Library, China National Knowledge Infrastructure, WanFang, and Sinomed databases for observational studies published between 1 January 2011 and 1 October 2021. Data were extracted using a standardized form to estimate the pooled vaccination coverage rate and 95% confidence intervals (CI) based on inclusion and exclusion criteria. Subgroup analysis was employed to explore heterogeneity. This study is registered in PROSPERO, CRD42021293175. Results: We identified 5128 records, of which 21 articles that included 34,6571 adults. The pooled coverage rate and 95% confidence intervals were 26.27% and 22.73–29.82%, respectively. The pooled coverage rates were 22.06% (95% CI: 15.35–28.78%), 33.81% (95% CI: 28.77–38.85%) and 23.50% (95% CI: 17.37–29.64%) in eastern China, central China and western China, respectively. Furthermore, males had a pooled hepatitis B vaccination coverage rate of 23.47% (95% CI: 15.61–31.33%), whereas, in females, the coverage rate was 26.60% (95% CI: 18.73–34.47%). The pooled hepatitis B vaccination coverage rate in the age group younger than 40 years was 36.93% (95% CI: 28.35–45.50%), while in the ≥40-year-old group, the pooled hepatitis B vaccination coverage rate was 17.09% (95% CI: 10.18–24.00%). The pooled hepatitis B vaccination coverage rate in urban areas (40.29%, 95% CI: 20.91–59.67%) was higher than in rural areas (16.54%, 95% CI: 7.80–25.29%). The average weighted, pooled hepatitis B vaccination coverage rate was 26.53% (20.25–32.81%) in 2011–2015 and 26.12% (22.04–30.20%) in 2016–2021. Conclusions: This systematic review provides the hepatitis B vaccination coverage rate of adults in China (26.27%). The low prevalence of vaccine-mediated immunity among adults in China underscores the urgent need for targeted immunization strategies for vulnerable Chinese adults to ensure progress toward the target of eliminating hepatitis B by 2030.

## 1. Introduction

As a major public health issue worldwide, the hepatitis B virus (HBV) infected 296 million people in 2019 and causes 1.5 million new infections each year [1]. The global burden of disease study estimated that there were 550,000 deaths in 2019, with a global mortality rate of 57 per 100,000, due to diseases such as liver cancer and cirrhosis caused by hepatitis B [2,3]. The World Health Organization (WHO) published the first Global Health Sector Strategy on Viral Hepatitis 2016–2021 and proposed nine quantitative global targets that included ‘reducing new cases of chronic viral hepatitis B and C infections by 90% and deaths by 65% by 2030 [4]. The Chinese Center for Disease Control and Prevention stated that the incidence of reported acute hepatitis B in China declined from 7.52 per 100,000 to 3.21 per 100,000 from 2005 to 2019 [5], and the China National Health Commission reported that the incidence of hepatitis B in China declined from 820.32 per 100,000 to 69.25 per 100,000 from 2011 to 2021 [6]. A 2019 systematic review and meta-analysis estimated the prevalence of HBV in China was 5–7.99% from 2013 to 2017, and most of the infected population were adults aged 20 years and older [7].

Improving the hepatitis B vaccination coverage rate could reduce HBV infection, and efforts to increase the number of children vaccinated against hepatitis B have been made in China over the past three decades. Since 1992, the hepatitis B vaccine has been included in program immunization management; since 2002, it was included in the national child program immunization scope, and from 1992 to 2006, the number of hepatitis B virus infections in China decreased by nearly 80 million [8]. From 2009 to 2011, the Ministry of Health of the People’s Republic of China conducted a HepB catchup campaign for children <15 years of age who were born between 1994 and 2001 [9]. In addition, administration of the universal HepB vaccine to all infants, as well as an HBsAg screening service for pregnant women, have been implemented by the government to block mother-to-child transmission (MTCT) [10]. In 2015, the children’s hepatitis B vaccination coverage rate was 99.6%, achieving the WHO target (90%) [11].

In 2011, the Chinese Center for Disease Control and Prevention issued a technical guide for adult hepatitis B immunization in China to encourage adult hepatitis B vaccination [12]. Although adult it is believed that hepatitis B vaccination coverage should be enhanced in China [5], and a systematic review study showed that hepatitis B vaccination is cost-effective in adults [13], there is still a lack of nationwide evidence on the vaccination rate against hepatitis B in China.

Evaluating the hepatitis B vaccination coverage rate among Chinese adults will help the government monitor progress in achieving the goal of eliminating hepatitis B by 2030. This study aimed to systematically evaluate the hepatitis B vaccination rate of adults in China and explore adults’ vaccination coverage in different age groups, sexes, regions, and different years to provide evidence-based data for the evaluation of the progress in and the barriers to implementing the nation’s HBV immunization strategy.

## 2. Methods

### 2.1. Protocol Registration and Search Strategy

The research protocol was registered in the PROSPERO international prospective register of systematic reviews (number: CRD42021293175). We searched six databases, including international databases (PubMed, EMBASE, Cochrane Library) and national databases (China National Knowledge Infrastructure (CNKI), WanFang data, and SinoMed), limiting the search to human subjects and publications in English and Chinese. Articles were identified using the search terms (“Hepatitis B” and “vaccine” and (“coverage” or “uptake” or “prevalence” or “proportion” or “status”) and “humans”). The details of the search strategy are listed in Appendix A. The search was conducted from 15 December 2021 to 20 December 2021 by two authors (Xinxin Bai and Lu Chen) independently, and any conflicts and quality assessments were resolved through discussions with a senior investigator (Guangyu Hu).

Here, we report our results in accordance with the standards of Preferred Reporting Items for Systematic Reviews and Meta-Analyses guidelines [14]. The flow diagram of the systematic literature review was produced by the PRISMA2020 compliant flow diagram R package [15].

### 2.2. Inclusion and Exclusion Criteria

The inclusion criteria were as follows: (1) articles that included at least two of the following pieces of information (the total number of adults surveyed, the number of adults who were vaccinated (receipt of ≥1 dose)); (2) studies that were conducted in China; (3) studies with a cross-sectional design; (4) original research articles published in English or Chinese; and (5) all articles published from 1 January 2011 to 1 October 2021.

The exclusion criteria were as follows: (1) studies on a targeted population, including studies that were based on sex-specific subgroups and studies on occupational subgroups or clinical subgroups, because these groups are not representative of the general population; (2) studies with missing key data; (3) studies with duplicate data; (4) hospital-based research; and (5) studies with sample sizes of less than 500 people.

### 2.3. Data Extraction

Two authors (Xinyao Liu and Yujia Tong) independently extracted the information using a standardized form for each study, including study name, first author’s name, year of publication, geographical location, sex, age groups, urban or rural, sample size, vaccination cases, and vaccination coverage. To avoid missing any additional articles, the reviewer also checked the references in all identified full-text articles. Each of the two independent reviewers read the full-text articles and extracted the data. Any conflicts were resolved through consensus.

### 2.4. Quality Assessment

The quality of all included articles was assessed using a modified version of the Cross-Sectional/Prevalence Study Quality Assessment Forms, which were recommended by the Agency for Healthcare Research and Quality (AHRQ) [16]. The AHRQ forms and their modified versions applied in this research are shown in Appendix A. They assess the source of information, sample representativeness, inclusion and exclusion criteria, thoroughness of descriptive statistics reporting, definition of the results, quality assurance, and response rates. There are 10 questions, which were answered with “Yes”, “No”, and “Unclear”. “Yes” represented a score of 1, and “No” or “Unclear” represented a score of 0. In the meta-analysis, the total score of the 10 questions in the assessment forms was used to assess the quality of each full-text article. Studies were judged to be at high risk of bias (1–4 points), middle risk of bias (5–7 points), or low risk of bias (8–10 points). The quality of all articles was independently examined by two authors (Xinxin Bai and Xinyao Liu), and any discrepancies were resolved through discussion and adjudication.

### 2.5. Data Synthesis and Analysis

We calculated the pooled proportion of hepatitis B vaccine uptake and 95% confidence interval (CI) using the proportion of HBV vaccine uptake reported in each included study. Then, we conducted a meta-analysis to critically evaluate and quantitatively synthesize evidence across studies that was shown in a forest plot.

Heterogeneity was assessed using the methods of Higgins and Green using a χ^2^-test with n-1 degrees of freedom, with *p* < 0.05 indicating statistical significance [17]. The *I*^2^ test was used to assess the proportion of variation in vaccination coverage, with values of 25%, 50%, and 75% representing low, medium, and high levels of heterogeneity, respectively [18,19]. When appropriate, data were pooled using a random-effects (DerSimonian and Laird method) model to present a more conservative assessment of effect in the presence of heterogeneity [20]. Publication bias was assessed graphically by funnel plots and formally by the Egger’s test (significance at *p* < 0.05).

Subgroup analyses was conducted to explore variations in pooled vaccination coverage rate among regions, sexes, age groups, urban or rural areas, and investigation periods. Combining the results of multiple studies increases the statistical power and improves estimates. Statistical analysis was performed by Stata software (Version 16.0, Stata Corp, College Station, TX, USA).

## 3. Results

A total of 5128 articles were identified from the 6 databases. After removing duplicates, we screened the titles and abstracts of the remaining 2330 articles, and 2245 articles were excluded based on the exclusion criteria. A total of 85 articles were retrieved in full text and selected for full-text evaluation, thus selecting 21 articles with a total of 346,571 participants who met the inclusion criteria and were included in the analysis [21,22,23,24,25,26,27,28,29,30,31,32,33,34,35,36,37,38,39,40,41]. Of the relevant studies, 64 full-text articles were excluded for the following reasons: 38 reporetd incomplete data, 13 studies were conducted before 1 January 2011, 11 had sample sizes <500, and 2 studies did not have enough information. The flow diagram of the search process is shown in Figure 1.

### 3.1. Study Selection and Characteristics of the Included Studies

All 21 included studies were cross-sectional and reported hepatitis B vaccination coverage rates. Out of the 21 studies, 20 were published in Chinese, and 1 was published in English. The sample size was between 539 and 260,950. Out of the 21 studies, 10 were undertaken in eastern China, 4 were undertaken in central China, and 6 were undertaken in western China. The area (urban/rural) distribution was urban, six studies, and rural, six studies. A summary of the characteristics of each study is provided in Table 1. The vaccination coverage rate ascertained, defined by the proportion of vaccinated individuals, ranged from 7.19% to 55.38%.

### 3.2. Study Quality

We assessed the risk of bias for all studies. The modified AHRQ item score results for each study are shown in Appendix A. The quality assessment scores of the selected studies varied from 1 to 6, for a possible score of 10.

### 3.3. Pooled Proportion of Hepatitis B Vaccine Uptake

All studies (*n* = 21) quantified adults’ uptake of the hepatitis B vaccine. Figure 2 presents the forest plot of the coverage rate of hepatitis B vaccination among adults in China. The pooled coverage rate was 26.27% (95% CI: 22.73–29.82%), but there was high heterogeneity (*I*^2^ = 99.6%, *p* <0.001). The coverage rate reported by the individual studies ranged from 7.19% to 55.38%.

### 3.4. Subgroup Analysis

We conducted a series of prespecified subgroup analyses and summarized the results in Table 2.

The pooled estimated hepatitis B coverage rate showed significant differences in different regions. For eastern China, the hepatitis B coverage rate could be calculated in 10 studies, with a pooled coverage rate of 22.06% (95% CI: 15.35–28.78%; *I*^2^ = 99.6%). For central China, the hepatitis B coverage rate could be calculated in four studies, with a pooled coverage rate of 33.81% (95% CI: 28.77–38.85%; *I*^2^ = 95.7%). For western China, the hepatitis B vaccination coverage rate could be calculated in six studies, with a pooled coverage rate of 23.50% (95% CI: 17.37–29.64%; *I*^2^ = 99.4%).

Furthermore, in 10 studies, males had a pooled hepatitis B vaccination coverage rate of 23.47% (95% CI: 15.61–31.33%; *I*^2^ = 99.2%), whereas in females, the coverage rate was 26.60% (95% CI: 18.73–34.47%; *I*^2^ = 99.3%). The pooled hepatitis B vaccination coverage rate in the age group younger than 40 was 36.93% (95% CI: 28.35–45.50%; *I^2^* = 99.1%), while in the age group ≥40 years old, the pooled hepatitis B vaccination coverage rate was 17.09% (95% CI: 10.18–24.00%; *I*^2^ = 99.7%). The pooled hepatitis B vaccination coverage rate in urban areas (40.29%, 95% CI: 20.91–59.67%; *I*^2^ = 99.3%) was higher than that in rural areas (16.54%, 95% CI: 7.80–25.29%; *I*^2^ = 98.8%). The hepatitis B vaccination coverage rates from 2011 to 2015 and from 2016 to 2021 were reported in 8 and 13 studies, respectively. The average weighted, pooled hepatitis B vaccination coverage rate was 26.53% (20.25–32.81%) in 2011–2015 and 26.12% (22.04–30.20%) in 2016–2021.

### 3.5. Heterogeneity and Publication Bias

The results of the heterogeneity test indicated that the studies were significantly heterogeneous (*I*^2^ = 99.6%, *p* < 0.001). The presence of publication bias was tested by Egger’s test and shown graphically by a funnel plot. Visual inspection of the funnel plot of studies reporting on the hepatitis B vaccination coverage rate revealed significant asymmetry (*p* = 0.011) using the Egger test (Figure 3).

## 4. Discussion

To the best of our knowledge, this current study is the first comprehensive systematic review that aimed to pool the coverage rate of hepatitis B vaccination in China. The volume of survey results that have become available during the past decade was sufficient to produce intuitive and reasonable estimates, which adds to the novelty of our study. Based on data from 21 cross-sectional studies covering 346,571 individuals across all population groups, we found that the pooled coverage rate of the hepatitis B vaccine was 26.27% (95% CI: 22.73–29.82%). Our results suggest that fewer than 27 in 100 adults had been vaccinated during the past decade, from 2011 to 2021.

China has implemented a series of projects to achieve targets for hepatitis B vaccination coverage for children and infants and for the prevention of mother-to-child transmission of HBV [42]. However, no catch-up vaccination policy has been implemented nationwide for the adult population. A national health interview survey showed that the hepatitis B vaccination coverage rate of adults in the United States was 25.0% (95% CI: 24.3–25.8%) in 2013 [43]. In Germany, hepatitis B vaccination coverage in adults was below the 95% goal defined by the WHO, because catch-up vaccinations are currently only recommended for selected populations of adults according to a systematic review published in 2021 [44]. In 2015, Lu et al. [45] estimated that the hepatitis B vaccination coverage among adults in the U.S.A. traveling to a country of high or intermediate endemicity was 38.6%. Despite China being a high endemic area of HBV infection and having taken measures to improve the vaccination rate [46], the adult vaccination coverage is lower than other countries, which warrants further attention.

In addition, this study found that the coverage rate of the hepatitis B vaccine in China varied across provinces; it was lower in the eastern region than in other regions and was significantly lower in rural than urban settings. This result is inconsistent with previously published evidence by Tan et al. [47]. The level of development of the eastern region and urban areas is relatively high in China. No relationship between low socioeconomic level and vaccination coverage rate was found in our study, which might be due to the limited number of studies included in the meta-analysis. According to the National Bureau of Statistics of China, in 2021, the proportion of people over 40 years was 54.97% in rural areas and 45.45% in urban areas; the population ratios would partly explain the difference in vaccine coverage between rural and urban areas [48].

A significant vaccination coverage difference by gender was found in our study. The pooled coverage rate of hepatitis B vaccination in males was 23.47% (95% CI: 15.61–31.33%), while in females it was 26.60% (95% CI: 18.73–34.47%). This is consistent with a national health interview survey in the U.S.A. that showed that the hepatitis B vaccination coverage rate in females is higher than in males in high-risk populations [49], which may be associated with greater risk perception in females.

For the variation among different age groups, we found that the pooled coverage rate of hepatitis B vaccination for those younger than 40 years was higher than for those older than 40 years (36.93% (95% CI: 28.35–45.50%) vs. 17.09% (95% CI: 10.18–24.00%)). Meanwhile, the prevalence of HBV infection in the 20–39-year-old population was higher than in other age groups in China [6]. The Chinese government implemented a series of policies for the population of childbearing age to reduce HBV infection, including providing a free hepatitis B vaccine, which may account for the younger population’s higher coverage rate [10]. In addition, there was no significant change in the hepatitis B vaccination coverage rate among studies conducted between 2011 and 2015 and between 2016 and 2021 in our meta-analysis. This highlights the need for effective strategies to enhance adult vaccination coverage in China in the future.

Although China has made progress in blocking HBV MTCT through collaboration between maternal and child healthcare delivery systems and governments [10], the different policies for adult hepatitis B vaccination by the local governments pose new challenges to China’s HBV elimination [8]. By providing government financial subsidies, Beijing, Shanghai, and other metropolis have provided free hepatitis B vaccines to adults, while other less developed regions have implemented a co-pay or pay-out-of-pocket policy [50]. According to the Technical Guide for Adult Hepatitis B Immunization in China published by the Chinese CDC, hepatitis B vaccination is recommended for all adults over 18 years of age who have not received the hepatitis B vaccine, especially those at risk of sexually exposed infection, those at risk of occupational exposure, and those at risk of blood exposure via skin and mucous membranes [12]. These call for community engagement to jointly promote the adult immunization strategy. The engagement of private sector providers in immunization in the Western Pacific region has been proved to be beneficial to improve the overall efficiency of immunization services delivery [51].

The study’s limitations should be acknowledged. First, despite the fact that the included articles were assessed for quality, the precision of the pooled effect size estimates was reduced due to heterogeneity between studies linked to the large number of studies included and the various study settings. Second, due to the studies with samples size less than 500 being excluded in the review, the results with potential bias should be interpreted with caution. Third, few studies reported the hepatitis B vaccination coverage rate at a national level. This scarcity should be addressed in further research, and continued monitoring of vaccination coverage is essential. Fourth, due to the cross-sectional design of the included studies, the reported factors associated with the hepatitis B vaccination coverage rate have heterogeneity and coverage bias, and the power of synthesizing the estimated hepatitis B vaccination coverage rate should be noted. Fifth, the protection rate and cost-effectiveness of actual vaccines should also be a concern. Janssen and colleagues proposed combining seroprotection rates with program completion rates to estimate the effective vaccine protection rates, which would be a meaningful method to assess real-world cost-effectiveness by evaluating cost-per-protected patients [52].

## 5. Conclusions

This systematic review provides evidence of the hepatitis B vaccination coverage rate of adults in China (26.27%). The low prevalence of vaccine-mediated immunity among adults in China underscores the urgent need for targeted immunization strategies for vulnerable Chinese adults to ensure progress toward the target of eliminating hepatitis B by 2030.

## Figures and Tables

**Figure 1 vaccines-10-00900-f001:**
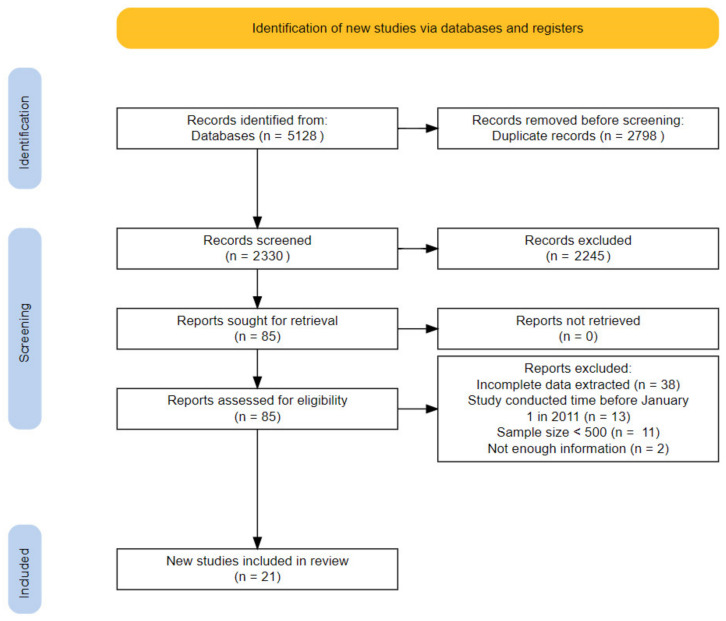
Flow diagram of the study selection process.

**Figure 2 vaccines-10-00900-f002:**
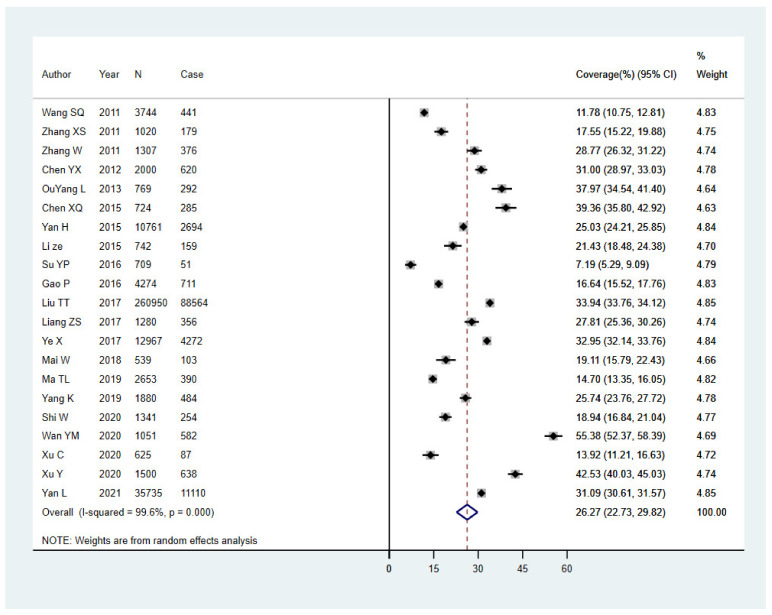
Forest plot of the 21 studies included in the meta-analysis.

**Figure 3 vaccines-10-00900-f003:**
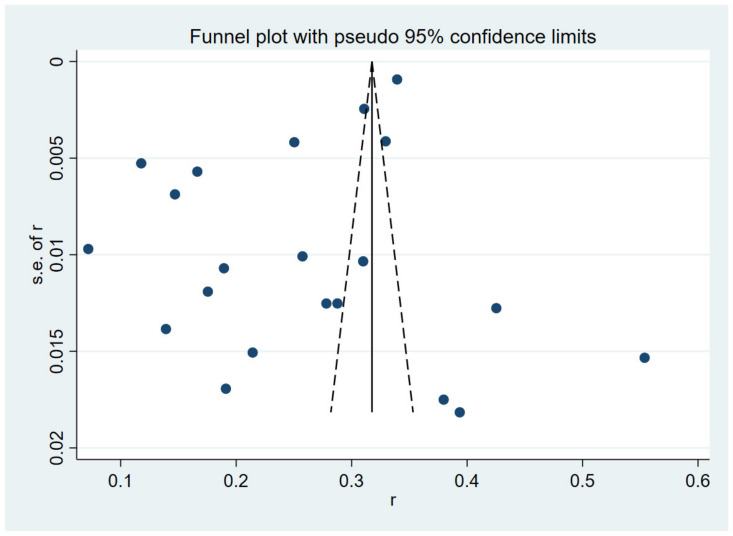
Funnel plot of the 21 studies included in the meta-analysis.

**Table 1 vaccines-10-00900-t001:** Characteristics of the included studies.

Author	Year of Publication	Location	Sample Size (N)	No. Vaccinated	Coverage Rate (%)	Age, y	Male, No. (%)	Quality Score	Risk of Bias Assessment
Wang SQ et al.	2011	Nanjing City	3744	441	11.78	Mean (SD):47.68 (17.36)	1632 (43.59)	4	high
Zhang XS et al.	2011	Gansu Province	1020	179	17.55	Range: 20–59	NA ^1^	4	high
Zhang W et al.	2011	Lanzhou City	1307	376	28.77	Range: 20–59	NA	6	middle
Chen YX	2012	Xuzhou City	2000	620	31.00	Range: 21–60	1061 (53.05)	4	high
OuYang L	2013	Changsha City	769	292	37.97	Range: 18–50	NA	5	middle
Chen XQ	2015	Changzhi City	724	285	39.36	Range: 18–55	401 (55.39)	6	middle
Yan H et al.	2015	Qiqihar City	10,761	2694	25.03	NA	5576 (51.82)	6	middle
Li ze et al.	2015	Dali City	742	159	21.43	NA	NA	1	high
Su YP et al.	2016	Beijing	709	51	7.19	Mean (SD):44.3 (13.4)	311 (43.86)	5	middle
Gao P et al.	2016	Beijing	4274	711	16.64	NA	NA	5	middle
Liu TT et al.	2017	Mianyang City	260,950	88,564	33.94	Mean (SD):52.96 (15.34)	113,184 (43.37)	6	middle
Liang ZS	2017	Yancheng City	1280	356	27.81	NA	695 (54.30)	4	high
Ye X et al.	2017	Ganzhou City	12,967	4272	32.95	Mean (SD):54.64 (14.15)	7560 (58.30)	6	middle
Mai W et al.	2018	Zhaoqing City	539	103	19.11	Range: 24–59	206 (38.22)	5	middle
Ma TL et al.	2019	Hebei Province	2653	390	14.70	Mean (SD):40.9 (11.6)	1221 (46.02)	5	middle
Yang K et al.	2019	Ma’anshan City	1880	484	25.74	Mean: 42	713 (37.93)	6	middle
Shi W et al.	2020	Zhejiang Province	1341	254	18.94	Range: 30–59	NA	5	middle
Wan YM et al.	2020	Heilongjiang and Gansu Province	1051	582	55.38	Mean (SD):36.93 (10.35)	429 (40.82)	6	middle
Xu C	2020	Yulin City	625	87	13.92	Range: 18–65	268 (42.88)	6	middle
Xu Y et al.	2020	Zhejiang Province	1500	638	42.53	Mean (SD):36.78 (12.3)	715 (47.67)	6	middle
Yan L et al.	2021	Beijing	35,735	11,110	31.09	Range: 20–100	13,464 (37.68)	6	middle

^1^ NA: Not available.

**Table 2 vaccines-10-00900-t002:** Subgroup meta-analysis of studies reporting hepatitis B vaccination coverage in China from 2011–2021.

Subgroup	Number of Studies	Coverage (%)	95% CI	*I*^2^ (%)	Heterogeneous *p*-Value	Egger’s Test *p*-Value
Region						
Eastern	10	22.06	15.35–28.78	99.6	*p* < 0.001	0.047
Middle	4	33.81	28.77–38.85	95.7	*p* < 0.001	0.906 *
Western	6	23.50	17.37–29.64	99.4	*p* < 0.001	0.026 *
Gender						
Male	10	23.47	15.61–31.33	99.2	*p* < 0.001	0.161
Female	9	26.60	18.73–34.47	99.3	*p* < 0.001	0.387 *
Age group						
<40	10	36.93	28.35–45.50	99.1	*p* < 0.001	0.005
≥40	10	17.09	10.18–24.00	99.7	*p* < 0.001	0.062
Urban/Rural						
Urban	6	40.29	20.91–59.67	99.3	*p* < 0.001	0.114 *
Rural	6	16.54	7.80–25.29	98.8	*p* < 0.001	0.034 *
Investigation period						
2011–2015	8	26.53	20.25–32.81	99.0	*p* < 0.001	0.906 *
2016–2021	13	26.12	22.04–30.20	99.6	*p* < 0.001	0.069

*: When the sample size is less than 10 articles, it makes little sense to conduct an Egger’s test.

## Data Availability

Data sharing is not applicable for this systematic review and meta-analysis. Data used in this study are available from the included published papers.

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
