# Peer review of "Adult Hepatitis B Virus Vaccination Coverage in China from 2011 to 2021: A Systematic Review"

_vaccines, 2022, doi:10.3390/vaccines10060900_

Round 1

Reviewer 1 Report

The study investigated HBV-vaccine coverage rate of adults in China with extensive systematic review and showed 26.27% of coverage rate during recent 10 years. The submitted manuscript was well organized and written. I'd like to give a few suggestions to make the study more brilliant.   Please, consider to add incidence of acute hepatitis B among Chinese adults during recent 10 years in the Introduction. If possible, it'd be better to show recent trend of acute hepatitis B among adults in China.   Line 92: what is the definition of vaccine coverage? Three times or at least one injection of a vaccine?   Line 94: Why did you select 2011 as a begining of the study? Was there any national policy change againt HBV since 2011?   Line 240-252: The difference of vaccine coverage among the provinces and between rural and urban areas might be associated with different age groups. The more old age group might affect the coverage rate in the rural area. Please, add some comment on the possible effect of different age groups on the vaccine coverage in them.

Author Response

Point 1: Please, consider to add incidence of acute hepatitis B among Chinese adults during recent 10 years in the Introduction. If possible, it'd be better to show recent trend of acute hepatitis B among adults in China.

Response 1: Thanks for your comments. We revised the section in paragraph 1 of the introduction as follows: “ The Chinese Center for Disease Control and Prevention reported the incidence of reported acute hepatitis B in China declined from 7.52 per 100,000 to 3.21 per 100,000 from 2005 to 2019[5], and China National Health Commission reported the incidence of hepatitis B in China declined from 820.32 per 100,000 to 69.25 per 100,000 from 2011 to 2021[6]. While a most recent systematic review estimated the prevalence of HBV in China was still 5%–7.99% from 2013 to 2017, and more than 90% of the HBV-infected population were adults[7].”

Point 2: Line 92: what is the definition of vaccine coverage? Three times or at least one injection of a vaccine?

Response 2: We revised the section as follows:

2.2. Inclusion and exclusion criteria --“The inclusion criteria were as follows: (1) articles that included at least two of these data (the total number of adults surveyed, the number of adults who were vaccinated, and vaccination coverage (received at least 1 dose of hepatitis B vaccine))”

Point 3: Line 94: Why did you select 2011 as a begining of the study? Was there any national policy change againt HBV since 2011?

Response 3: We revised the section as follows:

Introduction – paragraph2- “During 2009 – 2011, the Ministry of Health of the People’s Republic of China conducted a HepB catchup campaign for children <15 years of age who were born during 1994 – 2001[9].”

Introduction – paragraph3 - “In 2011, the Chinese Center For Disease Control And Prevention issued a technical guide for adult hepatitis B immunization in China to encourage adult hepatitis B vaccination[12].”

Point 4: Line 240-252: The difference of vaccine coverage among the provinces and between rural and urban areas might be associated with different age groups. The more old age group might affect the coverage rate in the rural area. Please, add some comment on the possible effect of different age groups on the vaccine coverage in them.

Response 4: We revised the section as follows: “According to National Bureau of Statistics of China, the proportion of people over 40 years old in rural areas is 54.97% and 45.45% in urban areas in 2021, the population ratio would partly explain the difference in vaccine coverage between rural and urban areas[48].”

Reviewer 2 Report

The manuscript is a review about the data published about the degree of hepatitis B vaccination in China.

The results give an insight of the situation and could contribute in the way of taking steps for improving the vaccination degree.

Somethings to improve in the explanation in the text and in Figure 1 it is no clear to what the authors refer by works not retrieved.

It is not also clear why 2245 publications were excluded.

Author Response

Point 1: Somethings to improve in the explanation in the text and in Figure 1 it is no clear to what the authors refer by works not retrieved.

Response 1: Thank you for the comments and we retrieved all 85 articles and Figure 1 had been replaced.  We revised the section as follows: “A total of 85 articles were retrieved in full text and selected for full-text evaluation…” in lines 154-155.

Point 2: It is not also clear why 2245 publications were excluded.

Response 2: Thank you for the comments, we revised the section as follows:

“we screened the titles and abstracts of the remaining 2,330 articles, and 2245 articles were excluded based on the exclusion criteria”

Reviewer 3 Report

The manuscript by Bai et al. “Adult hepatitis B virus vaccination coverage in China from 2011 2 to 2021: A systematic review” describes a metanalysis of studies assessing HBV vaccination coverage in China. The authors declare that their main concern during the screening of literature articles was to avoid biases, and therefore they deliberately excluded subpopulation studies and studies including less than 500 subjects. However, by doing this, they possibly missed the point, which is what they themselves declared in the discussion: to provide useful data for guiding decisions in public health policies. In my opinion, the most reliable way to achieve this fundamental objective is analyzing how HBV vaccination policies (and prophylactic measures against vertical transmission, that are also of the utmost importance) are being implemented and how they impact in newborn and children, rather than providing an aseptic view of the whole population. In addition, there is no trace of any analysis performed on the pregnant women population. This population is doubly important, 1) because it is most often a reliably unbiased population and 2) because of its importance in vertical transmission. Therefore, a more focused analysis on the younger generations, both including dedicated studies and better extrapolating data from the ones the authors have selected, would have provided this manuscript with greater scientific interest, which otherwise appears, despite the scientific rigor of the authors, somewhat scholastic and basically superficial as it doesn’t tackle the important points. As the matter is instead crucial and serves an important function for public health issues in China, the authors should radically amend their study (and not just the manuscript) according to the following points:

  • include subpopulation studies analyzing newborns and children
  • include studies analyzing pregnant women
  • extrapolate these populations from the general population
  • elaborate specific analyses and include specific figures displaying vaccination rate time trends in newborns and infants in different geographic areas
  • include data and specific figures on the penetration of HBsAg screening in pregnant women and of the prophylactic measures implemented to prevent vertical transmission.

Author Response

Point 1: The authors declare that their main concern during the screening of literature articles was to avoid biases, and therefore they deliberately excluded subpopulation studies and studies including less than 500 subjects. However, by doing this, they possibly missed the point, which is what they themselves declared in the discussion: to provide useful data for guiding decisions in public health policies.

Response 1: Thanks for the comment. We revised the limitation section as follows:

“The limitations of this study should be acknowledged. First, although strict selection and quality assessment were made in the included articles, heterogeneity between studies was high owing to the number of studies included and the various study regions and publication years, which reduced the precision of our pooled effect size estimates. Thus, subgroup analysis was used to minimize this heterogeneity and provided more details on hepatitis B vaccination. Second, due to the studies with sample size less than 500 being excluded in the review, the result with potential bias should be interpreted with caution. Third, few studies reported the hepatitis B vaccination coverage rate at a national level. This scarcity should be addressed in further research and continued monitoring of vaccination coverage is essential. Fourth, the reported factors associated with the hepatitis B vaccination coverage rate had heterogeneity and coverage bias due to the cross-sectional design of the included studies and the limitation of the power of synthesizing the estimated hepatitis B vaccination coverage rate should be noted. Fifth, the protection rate and cost-effectiveness of actual vaccines also should be a concern. Janssen and colleagues proposed combining seroprotection rates with program completion rates to estimate effective vaccine protection rates, which would be a meaningful method to assess real-world cost-effectiveness by assessing cost-per-protected patient[52].”

Point 2: In my opinion, the most reliable way to achieve this fundamental objective is analyzing how HBV vaccination policies (and prophylactic measures against vertical transmission, that are also of the utmost importance) are being implemented and how they impact in newborn and children, rather than providing an aseptic view of the whole population.

Response 2: Thank you for your comments, we revised the section as follows:

Introduction - paragraph 2-3:

“Improving the hepatitis B vaccination coverage rate could reduce HBV infection, and efforts to increase the number of children vaccinated against hepatitis B have been made in China over the past three decades. Since 1992, the hepatitis B vaccine has been included in program immunization management; since 2002, it was included in the national child program immunization scope, and from 1992 to 2006, the number of hepatitis B virus infections in China decreased by nearly 80 million[8]. During 2009 – 2011, the Ministry of Health of the People’s Republic of China conducted a HepB catchup campaign for children <15 years of age who were born during 1994 – 2001[9]. While the universal HepB for all infants, HBsAg screening among pregnant women, and timely HBIG plus birth-dose of HepB for neonates with HBsAg positive mothers are used to prevent HBV mother-to-child transmission (MTCT)[10]. In 2015, the children’s hepatitis B vaccination coverage rate was 99.6%, achieving the WHO target (90%)[11].”

“In 2011, the Chinese Center For Disease Control And Prevention issued a technical guide for adult hepatitis B immunization in China to encourage adult hepatitis B vaccination[12]. Although adult hepatitis B vaccination coverage is considered should be enhanced in China[5] and a systematic review study showed that hepatitis B vaccination is cost-effective in adults[13], there is still a lack of nationwide evidence on the vaccination rate of the hepatitis B vaccine in China.”

Point 3: In addition, there is no trace of any analysis performed on the pregnant women population.  This population is doubly important, 1) because it is most often a reliably unbiased population and 2) because of its importance in vertical transmission. Therefore, a more focused analysis on the younger generations, both including dedicated studies and better extrapolating data from the ones the authors have selected, would have provided this manuscript with greater scientific interest, which otherwise appears, despite the scientific rigor of the authors, somewhat scholastic and basically superficial as it doesn’t tackle the important points.

Response 3:  We appreciate the reviewer’s insightful suggestion. In the discussion section, we introduced the profile of prophylactic measures against vertical transmission in China and enlightenment for the adult vaccination strategy as follow:

Since 1992, the Chinese government has been working to prevent and control HBV transmission. To prevent HBV mother-to-child transmission (MTCT) in the pregnant women population, pregnant women are screened for HbsAg nationally and infants whose mothers are HbsAg positive would be given a timely birthdose of HepB and additional hepatitis B immunoglobulin (HBIG) within 24 hours of birth (or as soon as possible after delivery). Two additional HepB doses are given in strict accordance with the national immunization schedule. Meanwhile, antiviral therapy can be given to HBV-infected pregnant women in their third trimester to reduce the risk of HBV MTCT. Additionally, routine childhood immunization as part of a universal HepB immunization plan was implemented to help reduce HBsAg prevalence among presently pregnant women. 

As a result of the strengthened prophylactic measures against vertical transmission, China national coverage of infants with the third dosage of HepB increased from 30% in 1992 to 99.58 % in 2015. Based on these successful experience, the elimination of MTCT of HIV, Syphilis, and HBV program (EMTCT) in 2016 was proposed and implemented as follow. In 2017, the national coverage of HBsAg screening test among pregnant women exceeded 99.5% and nearly all (99.7%) infants exposed to HBV received HBIG at birth.

Discussion-paragraph 6:

“The Chinese government has made considerable progress in preventing the MTCT of HBV in the past three decades based on close collaboration between maternal and child healthcare institutions and immunization departments[10]. However, China still faces challenges in eliminating HBV in adults, due to the local governments having different policies for hepatitis B vaccination for adults[8]. Some regions, such as Beijing, Shanghai, and Shaanxi, have implemented free hepatitis B vaccines for adults through government financial subsidies, while other regions still require people to pay out of pocket[49]. According to the Technical Guide for Adult Hepatitis B Immunization in China published by the Chinese CDC, hepatitis B vaccination is recommended for all adults over 18 years of age who have not received hepatitis B vaccine, especially those at risk of sexually exposed infection, those at risk of occupational exposure, and those at risk of blood exposure via skin and mucous membranes[12]. These call for community engagement to jointly promote the adult immunization strategy. And the engagement of private sector providers in immunization in the Western Pacific region has been proved to be beneficial to improve the overall efficiency of immunization services delivery[51].”